# Nitrogen-Doped Bismuth Nanosheet as an Efficient Electrocatalyst to CO_2_ Reduction for Production of Formate

**DOI:** 10.3390/ijms232214485

**Published:** 2022-11-21

**Authors:** Sanxiu Li, Yufei Kang, Chenyang Mo, Yage Peng, Haijun Ma, Juan Peng

**Affiliations:** 1State Key Laboratory of High-Efficiency Utilization of Coal and Green Chemical Engineering, College of Chemistry and Chemical Engineering, Ningxia University, Yinchuan 750021, China; 2Key Laboratory of Ministry of Education for Protection and Utilization of Special Biological Resources in Western China, School of Life Sciences, Ningxia University, Yinchuan 750021, China

**Keywords:** electrocatalysis, electronic structure, flow cell, large current density

## Abstract

Electrochemical CO_2_ reduction (CO_2_RR) to produce high value-added chemicals or fuels is a promising technology to address the greenhouse effect and energy challenges. Formate is a desirable product of CO_2_RR with great economic value. Here, nitrogen-doped bismuth nanosheets (N-BiNSs) were prepared by a facile one-step method. The N-BiNSs were used as efficient electrocatalysts for CO_2_RR with selective formate production. The N-BiNSs exhibited a high formate Faradic efficiency (FE_formate)_ of 95.25% at −0.95 V (vs. RHE) with a stable current density of 33.63 mA cm^−2^ in 0.5 M KHCO_3_. Moreover, the N-BiNSs for CO_2_RR yielded a large current density (300 mA cm^−2^) for formate production in a flow-cell measurement, achieving the commercial requirement. The FE_formate_ of 90% can maintain stability for 14 h of electrolysis. Nitrogen doping could induce charge transfer from the N atom to the Bi atom, thus modulating the electronic structure of N-Bi nanosheets. DFT results demonstrated the N-BiNSs reduced the adsorption energy of the *OCHO intermediate and promoted the mass transfer of charges, thereby improving the CO_2_RR with high FE_formate_. This study provides a valuable strategy to enhance the catalytic performance of bismuth-based catalysts for CO_2_RR by using a nitrogen-doping strategy.

## 1. Introduction

With rapid economic development and the extensive use of fossil fuels, carbon dioxide (CO_2_) in the atmosphere continues to increase. Excessive amounts of CO_2_ emissions will cause severe global warming and sea levels to rise. Therefore, the necessary measures to reduce the environmental impact of CO_2_ are of great importance. Among the various existing CO_2_ conversion technologies, the electrochemical CO_2_ reduction reaction (CO_2_RR) is a prospective strategy to produce fuels or value-added chemicals. Electrochemical CO_2_RR can convert CO_2_ into CO [1], formate [2], methanol [3], ethanol [4,5], and hydrocarbons [6,7,8] under room temperature and atmospheric pressure. Among the CO_2_ reduction products, formate has attracted great interest because of its commercial value. Mainly, formate is an essential feedstock for the pharmaceutical and chemical industries. Therefore, it is significant to develop an efficient way to produce formate.

The practical application of electrochemical CO_2_RR to formate production is constrained owing to the following factors, such as high overpotential, colossal cost, low selectivity, and poor stability [9]. In addition, since electrochemical CO_2_RR usually occurs in an aqueous solution, it will be accompanied by a hydrogen evolution reaction (HER), resulting in low Faradaic efficiency (FE) of the product. So far, many metal-based catalysts, including Pb [10], In [11], Sb [12,13], Sn [14], Co [15], Cu [16,17], Bi [18], and Pd [19] were designed and developed to improve the efficiency and selectivity of CO_2_RR to produce the formate in an aqueous solution. Bi-based catalysts are popular because of the selective production of formate, low cost, and ability to inhibit the occurrence of adverse reaction HER in an aqueous solution [20]. In addition, Bi-based catalysts tend to stabilize *OCHO, an essential intermediate in formate formation [21]. However, the catalytic efficiency and selectivity of Bi-based nanomaterials are far from meeting the requirements of CO_2_RR [22]. Therefore, there is an urgent need to develop effective and straightforward pathways to improve catalytic efficiency and selectivity. As demonstrated recently, plenty of studies have exhibited that the properties of CO_2_RR were improved by changing the structure of the catalyst, such as size and morphology [23,24,25]. Two-dimensional (2D) nanostructures with high conductivity and abundant active sites have been extensively studied as potential electrocatalysts, which hold a key to enhanced CO_2_RR activity [26,27]. Meanwhile, the FE of target products was improved via surface modification, doping hetero-atoms, and defects [28,29]. Especially for doping hetero-atoms, such as N, S, or O, this tactic would optimize the electronic structure of the catalyst and provide more active sites. For instance, Wu et al. [30] used this strategy to prepare the nitrogen-doped Sn(S) nanosheets. It can reach the highest FE of formate of 93.3% at −0.7 V versus reversible hydrogen electrode (vs. RHE). Liu et al. [31] prepared a Bi_2_O_3_-NGQDs catalyst and achieved the FE of formate was nearly 100% at −0.71 V (vs. RHE). Besides, Kang et al. [32] prepared nitrogen-doped SnO_2_/C material that could enhance electrocatalytic CO_2_RR activity, showing high FE (90%) for formate at –0.65 V vs. RHE. Therefore, doping hetero-atoms can increase the amounts of active sites and optimize the electronic structure to achieve high formate current density and selectivity.

Herein, we developed a one-step activation and nitrogen-doping combination method using bismuth citrate as the Bi precursor, and the combination of Ca(OH)_2_ and NH_4_Cl as an activator and a nitrogen source to prepare the nitrogen-doped bismuth nanosheets (N-BiNSs). The N-BiNSs can be used to electrocatalyze CO_2_RR to produce formate with high efficiency. N-doping improved the electrical conductivity, which will make up for the poor conductivity of bismuth-based materials. The theoretical study shows that the outstanding selectivity could be attributed to the change in the electronic structure of bismuth after doping N atoms. At the same time, N-BiNSs reduced the adsorption energy of *OCHO intermediate and promoted the mass transfer of charge. The optimal adsorption *OCHO intermediate promoted formate formation while inhibiting the CO product pathway, thereby enhancing the selectivity of CO_2_RR for formate.

## 2. Results and Discussion

### 2.1. Morphology and Structure Analysis

In this work, nitrogen-doped bismuth nanosheets (N-BiNSs) were prepared by a simple and rapid one-pot method. A series of N-BiNSs-X was successfully constructed by adjusting the amount of Ca(OH)_2_ and NH_4_Cl in different proportions to provide an activating agent and a nitrogen source (Figure 1). To study the phase composition of N-BiNSs, the catalyst was determined by X-ray diffraction (XRD) measurement. It can be drawn from the XRD pattern in Figure 2a, the N-BiNSs exhibited different peaks at 27.12°, 37.95°, and 39.62°, which were indexed to the (012), (104), and (110) planes of tripartite crystal system Bi (PDF#85-1329). Compared with BiNSs catalyst, the N-BiNSs had stronger (110) and (012) crystal planes. The morphologies of the BiNSs and N-BiNSs were studied by scanning electron microscopy (SEM), as shown in Appendix A and Figure 2b. It can be seen that the BiNSs showed the stacked flakiness morphology of bulk Bi (Appendix A), while the N-BiNSs showed a uniform 2D nanosheet structure with a large thin layer area (Figure 2b). This 2D nanostructure was very beneficial for increasing surface area and abundant active sites. Additionally, the transmission electron microscopy (TEM) observations (Figure 2c) revealed the ultrathin feature of N-BiNSs. These nanosheets were ubiquitously present. Besides, the high-resolution transmission microscopic (HRTEM) images of N-BiNSs display a lattice spacing of about 0.272 nm that corresponds to the (110) crystal plane of the tripartite crystal system Bi. The lattice band gap of the material was very uniform. As shown in Figure 2e, the high-angle annular dark field scanning TEM (HAADF-STEM) images also exhibited the morphology of nanosheets. Moreover, the N-BiNSs structure was confirmed again by the dispersive energy X-ray (EDX) elemental mapping. The Bi (red) and N (green) elements were well distributed on the whole surface of N-BiNSs (Figure 2f–h). Meanwhile, the EDS elemental mapping and line scan (Appendix A) were first recorded to demonstrate Bi and N distribution on N-Bi nanosheets accompanied by a disparate atomic ratio of 97.85% (Bi) and 2.85% (N). The above results suggested that nitrogen had been triumphantly incorporated into the BiNSs catalyst.

X-ray photoelectron spectroscopy (XPS) was also used to characterize the surface chemical constituents of the specimens and the valence state of the nanomaterials. The position of the C(sp2) peak in the C1s spectrum was taken as the reference value of 284.8 eV, and the obtained XPS spectrum was calibrated. Figure 3a,b reveals the detailed Bi 4f spectra by comparing the XPS spectra of BiNSs and N-BiNSs. For the BiNSs catalyst, binding energies at 165 eV and 159.6 eV were belong to the Bi^3+^ 4f_5/2_ and 4f_7/2_ [23], respectively. However, binding energies for Bi^3+^ 4f_5/2_ and 4f_7/2_ in N-BiNSs shifted to 164.5 and 159.11 eV, respectively. These could be caused by the charge transfer from the N to the Bi atom, thus optimizing the electronic structure of N-BiNSs. Furthermore, the N1s peaks of N-BiNSs can be split into three peaks (Figure 3c), pyridinic-N (397.6 eV), N-oxidized (404.6 eV), and Nitrate (405.4 eV) [12,33,34,35], respectively. XPS analyses showed that the atomic ratio of N was 0.41% and pyridinic N accounted for the largest proportion, it indicated that the N element was doped successfully in BiNSs.

### 2.2. Electrocatalytic Perfomance

To estimate the CO_2_ reduction performance of BiNSs, N-BiNSs, N-BiNSs-1, and N-BiNSs-2, the catalytic reactions are applied in a proton exchange membrane (PEM) separated two-compartment cell. Linear sweep voltammetry (LSV) curves (Appendix A) of N-BiNSs were measured in both CO_2_-saturated and Ar-saturated 0.5 M KHCO_3_ electrolytes. The N-BiNSs catalyst exhibited a larger current in the CO_2-_saturated electrolyte than in N_2_. The current density increases sharply from −0.6 V (vs. RHE), reaching about 30 mA cm^−2^ at −0.8 V (vs. RHE) to about 50 mA cm^−2^ at −1.2 V (vs. RHE). The polarization curves stated that the catalyst has higher activity to CO_2_RR. Meanwhile, the LSV curves (Figure 4a) were tested in the CO_2_-saturated 0.5 M KHCO_3_ electrolyte. Strikingly, for the N-BiNSs catalyst, the current density in the CO_2_-saturated atmosphere was higher than the BiNSs catalysts, indicating the introduction of N could improve the electrocatalytic activity of Bi on CO_2_RR. Moreover, comparing the three samples with different N doping amounts, the N-BiNSs show the highest current with the optimized Nitrogen amount of 2.85%.

To evaluate the selectivity of CO_2_RR, electrolysis was performed in a CO_2_-saturated 0.5 M KHCO_3_ aqueous solution at various applied potentials. The gaseous and liquid products were quantitatively analyzed via gas chromatography (GC) and ion chromatography (IC) (Appendix A), respectively. As shown in Figure 4b, the FE_formate_ of N-BiNSs was higher than BiNSs, N-BiNSs-1, and N-BiNSs-2 at all applied potentials. To investigate the catalytic activity of N-BiNSs, the CO_2_RR was carried out at different potentials between −0.4 V and −1.3 V (vs. RHE). As the applied potential changes, the FE of the CO_2_RR products was displayed in Figure 4c. The N-BiNSs also showed that formate was the main product of CO_2_RR. The FE value of CO was lower than 20% in the whole potential window, and the FE value of H_2_ decreased significantly from 60% at −0.5 V to 2–3% at −1.0 V (vs. RHE), the FE_formate_ at −0.95 V (vs. RHE) was 95.25%. In contrast, for the BiNSs without nitrogen element, as shown in Appendix A, the FE values of formate, CO, and H_2_ at −0.95 V (vs. RHE) were 81.54%, 3.09%, and 10.56%, respectively. This indicated that the nitrogen-doped bismuth catalyst was beneficial to improve the selectivity of formate. Meanwhile, in the N-BiNSs-1 catalyst with a small amount of nitrogen, the FE_formate_ could reach 88.94% at −0.95 V (vs. RHE) in Appendix A. For the catalyst N-BiNSs-2 with a higher amount of doping nitrogen, the maximum FE of formate in the H-shaped cell reached 78.63% in Appendix A. It could be concluded that varying N doping amounts lead to the formation of different catalyst morphologies, resulting in different activities. Appendix A showed the constant potential electrolysis of CO_2_ under a series of potentials. The stable current density showed that the N-BiNSs catalyst had good electrochemical stability in the CO_2_RR test. Furthermore, the formate partial current densities (j_formate_) of the N-BiNSs were measured in the whole potential region in Figure 4d, with a maximum value of j_formate_ = 45 mA cm^−2^ at −1.2 V (vs. RHE). To better understand the activity and kinetics of N-BiNSs materials for CO_2_RR, the Tafel diagram of the catalyst was obtained in the low current density area. In Appendix A, the slope value obtained by the N-BiNSs catalyst was 184.13 mV dec^−1^, indicating that the electron transfer rate of this catalyst was relatively fast, thus facilitating the adsorption and desorption of CO_2_* intermediate on the N-BiNSs catalyst surface [36].

More importantly, the CO_2_RR activity was also related to the electrochemical active surface area (ECSA) of the catalyst. To evaluate the ECSA of BiNSs, N-BiNSs, N-BiNSs-1, and N-BiNSs-2, the double-layer capacitance (Cdl) was calculated. According to the cyclic voltammograms (CVs) of BiNSs, N-BiNSs, N-BiNSs-1, and N-BiNSs-2 at different sweep speeds in the potential region of 0.12 V–0.22 V (vs. RHE) (Appendix A). It can be seen from Figure 4e and Appendix A that the capacitance values of BiNSs, N-BiNSs, N-BiNSs-1, and N-BiNSs-2 were 0.43 mF cm^−2^, 22.2 mF cm^−2^, 0.56 mF cm^−2^, and 0.075 mF cm^−2^, respectively. These results indicated that the Cdl of N-BiNSs electrocatalyst was highest, which can provide abundant catalytic active sites for increasing the electrocatalytic performance of CO_2_RR. At the same time, we measured the impedance of some different catalysts at open circuit voltage and obtained the Nyquist diagram (Appendix A) to explore the kinetic processes among the catalyst interfaces. The N-BiNSs material corresponds to the most minor semicircle among the three. The results showed that the interfacial charge could be transferred rapidly to improve the catalytic activity in the reaction process, which was consistent with our inferred results.

We further studied the long-term durability of the material for CO_2_RR, as shown in Figure 4f. The material was tested for electrolysis at −0.95 V (vs. RHE) for about 20 h, and the FE of formate remained very stable, basically around 95%, indicating that the N-BiNSs material had significant durability to CO_2_RR. A comparison of the performance of N-Bi nanosheets in electrocatalytic CO_2_RR with other representative electrocatalysts is in the literature (Appendix A). It was noteworthy that the N-BiNSs catalyst showed the morphology of the nanosheets was maintained and became thinner after a period of electrolysis process (Appendix A). The crystal shape of the nanosheets remained after prolonged electrolysis (Appendix A). It was proved that the catalyst had excellent morphology stability.

To eliminate CO_2_ mass transfer constraints in the H-cell and achieve a commercially viable high current density (≈200 mA cm^−2^), a flow cell reactor was assembled using catalysts coated on the gas diffusion layer, carbon paper of 2 × 3 cm size, and commercial mercury oxide anion exchange membrane (Appendix A). In this unit, carbon dioxide gas can react at the gas-liquid-solid three-phase boundary. Peristaltic pumps and gas-liquid mixed flow pumps are installed on the cathode and anode to remove liquid or gas products, keeping the pH of the electrolyte constant and fully contacting the electrode surface. Then, we systematically evaluated the CO_2_RR performance of the N-BiNSs catalyst on a carbon-based gas diffusion layer (C-GDL) substrate, as shown in Figure 5a,b. 1.0 M KOH aqueous solution was used as a flow-electrolyte. The alkaline electrolyte not only effectively inhibits HER, but also effectively reduces the activation energy barrier. Through the LSV curve, it can be displayed very clearly that the current density of the catalyst has met the commercial requirements, as shown in Figure 5a. Notably, the catalyst was maintained for more than 14 h at a high current density of ~300 mA cm^−2^ at a potential of −1.2 V (vs. RHE) in Figure 5b. The catalyst had significant durability at high current density, with the average selectivity of the catalyst being about 89.30%. After 14 h of electrolysis, due to the existing structure of the flow tank, flooding, seepage, and other problems occurred. Therefore, the optimization of the flow tank structure is still an urgent problem to be solved.

The two-electrode electrolyzer was assembled by utilizing the N-BiNSs loaded carbon paper as a cathode for CO_2_RR and IrO_2_ loaded on carbon paper as an anode for the oxygen evolution reaction (OER). As shown in Figure 5c, the polarization curve of the N-BiNSs||IrO_2_ cell exhibits electrocatalytic performance, with a current density of 6.19 mA cm^−2^ at 3.0 V, capable of delivering 10 mA cm^−2^ at 3.4V. It was worth noting that the initial current density was maintained at 6 mA cm^−2^. After 11 h of continuous reaction, the current density was maintained at 5.4 mA cm^−2^, the current density decreased by about 9.3%, and the stability was pretty, as shown in Figure 5d. The OER electrocatalyst IrO_2_ may also be replaced by non-noble metal-based materials [37].

### 2.3. Catalytic Mechanisms Revealed by DFT Computations

All the above electrochemical performance studies proved that the N-BiNSs catalysts had excellent activity and selectivity compared with pure BiNSs. In order to further explore the reasons why the catalyst improves CO_2_RR activity and selectivity, and determine the reaction mechanism of formate, the density functional theory (DFT) calculation was used to simulate and compare the CO_2_RR pathway on N-BiNSs and BiNSs surfaces. Figure 6a,b depicts the optimized adsorption geometries and their energy distributions for *OCHO (intermediate to formate) on N-BiNSs and BiNSs surfaces, respectively. As shown in Figure 6b, the total reaction pathway for the electrochemical reduction of CO_2_ to formate was two protons, and two electrons were transferred through the *OCHO intermediate and adsorbed by HCOOH (aq) [38]. Both N-BiNSs and BiNSs displayed a great energy barrier for *OCHO formation, confirming the original proton-coupled electron transfer is the potential limiting procedure [31] and that the optimally adsorbed *OCHO intermediate promotes formate production. Obviously, on the N-BiNSs (012) surface, the calculated Gibbs free energy ΔG for the formation of *OCHO was +0.36 eV, and the ΔG for the formation of *OCHO from BiNSs was +0.65 eV. Thus, the N-Bi nanosheets had a more negative Gibbs free energy than that of the Bi nanosheets site, indicating that *OCHO formation and protonation were more spontaneous.

## 3. Materials and Methods

### 3.1. Synthesis of Nitrogen-Doped Bismuth Nanosheets (N-BiNSs)

The synthesis method was improved according to the literature [39], and the synthesis of the specific process was as follows. Firstly, 5 g of C_6_H_5_BiO_7_, 3.7 g of Ca(OH)_2_, and 5.4 g of NH_4_Cl were weighed and placed in a sample vial the reagents were thoroughly mixed. Then the mixed reagents were transferred to a porcelain boat and placed in a tube furnace. Subsequently, the tube furnace was vacuumed, and N_2_ was pumped into it. The tube furnace was activated at 450 °C for 1 h in the N_2_ atmosphere, and then the temperature was increased to 800 °C for 2 h. During the whole synthesis process, the heating rate was 5 °C min^−1^ in the tube furnace. At the end of the reaction, the samples were cooled to room temperature, the samples were first repeatedly cleared with 2.0 M hydrochloric acid (HCl), and the inorganic salts of the samples were removed entirely. The samples were repeatedly cleared with deionized water until the pH value was neutral. Finally, the samples were dried under vacuum at 60 °C for 24 h. The samples were named nitrogen-doped bismuth nanosheets (N-BiNSs). For comparison, samples prepared at different Ca(OH)_2_ and NH_4_Cl were denoted as N-BiNSs-1 and N-BiNSs-2. The N-BiNSs-1 catalyst was prepared by adding Ca(OH)_2_ (1.85 g) and NH_4_Cl (2.7 g), and the N-BiNSs-2 was prepared by adding Ca(OH)_2_ (5.55 g) and NH_4_Cl (8.1 g). Moreover, the bismuth nanosheets (BiNSs) sample was prepared on the basis of the same procedure as introduced above, but without added Ca(OH)_2_ and NH_4_Cl.

### 3.2. Preparation of Working Electrode

Generally, 10 mg of as-prepared N-BiNSs-X were mixed with 600 μL of deionized water, 350 μL ethanol, and 50 μL of Nafion D-521 dispersion within a 2 mL vial in an ultrasonic bath for 2 h to become a homogeneous catalyst ink suspension; 10 μL of the catalyst ink was dropped onto a carbon paper (1 mg cm^−2^), and the catalyst-covered electrode was dried in a desiccator before use.

### 3.3. Electrochemical Measurements

In this work, all electrochemical properties were tested at an electrochemical workstation (Shanghai Chenhua, CHI760E). The electrocatalytic CO_2_RR was performed in an H-type electrolytic cell with the two chambers isolated by a Nafion 117 proton exchange membrane to stop reoxidation of the cathode generated to the anode. The prepared catalyst, the Ag/AgCl(saturated KCl), and the platinum sheet were used as the working electrodes, reference electrodes, and counter electrodes, respectively. All potential values were measured with Ag/AgCl and then converted to RHE. All electrode potentials were converted to electrode potentials relative to RHE by the Nernst equation: E (vs. RHE) = E (vs. Ag/AgCl) + 0.0591 × pH + 0.222 V. Each electrode chamber contained 35 mL of 0.5 M KHCO_3_ electrolyte. Before the test, the above electrolyte was continuously bubbled with high-purity CO_2_ or N_2_ for at least 30 min to saturate the electrolyte with N_2_ (pH = 8.5) or CO_2_ (pH = 7.2). The electrochemical reduction of CO_2_ on BiNSs and N-BiNSs-X electrodes were performed in N_2_-saturated 0.5 M KHCO_3_ or CO_2_-saturated 0.5 M KHCO_3_ at ambient temperature and pressure. Double-layer capacitance (Cdl) was obtained from cyclic voltammetry (CV) curves surveyed at unlike scan rates (10, 30, 50, 70, 90, 110, and 130 mV s^−1^) in the scope of +0.12 V to +0.22 V (vs. RHE). Electrochemical impedance spectroscopy (EIS) was employed to understand the charge mass transfer resistance in the electrocatalytic CO_2_RR course. The frequency range was 10 kHz–1.0 Hz and the amplitude was 5 mV s^−1^. The durability of the catalyst was obtained by using the current-time curve method (i-t). At the end of each potential test, the liquid products generated at each potential were collected and detected by ion chromatography.

### 3.4. Product Analysis

Faraday efficiency test method: Control potential electrolytic Coulomb method was used for the CO_2_ saturated solution, and the electrolytic reduction products were analyzed and calculated 0.5 h later. The flow rate of CO_2_ was mastered at 20 mL min^−1^ during electrolysis. The liquid products after electrolysis were detected by ion chromatography (AS-DV, Thermo Scientific, New York City, MA, USA). The Faraday efficiency (FE) of liquid-phase products was calculated as follows:$$\mathrm{FE}=\frac{NnF}{Q}\times 100\%$$
where *N* is the number of electrons transferred, *n* is the total mole fraction of the gas measured by gas chromatography, *F* is the Faraday constant (96,485 C mol^−1^), and *Q* is the total electric charge passed through the electrode.

The gaseous products were quantified by gas chromatography (GC7900, Tianmei, China), and the H_2_ and CO were detected by a thermal conductivity detector (TCD), and flame ionization detector (FID), respectively, and ultra-pure N_2_ (>99.99%) was used as carrier gas. The calculation method of Faraday efficiency (FE) was as follows:$$\mathrm{FE}=\frac{n\times C\times v\times F}{Vm\times j}\times 100\%$$
where *ν* is the gas flow rate of supplied CO_2_, *C* is the concentration of the gaseous product, *n* is the number of electrons for producing a molecule of CO or H_2_, *F* is the Faraday constant (96,485 C mol^−1^), *Vm* is the molar volume of gas at 298 K, *j* is the recorded current (A).

## 4. Conclusions

In summary, we successfully prepared nitrogen-doped bismuth nanosheets (N-BiNSs) through a simple one-step activation and nitrogen-doping connective way. The N-Bi nanosheets exhibited very high activity, selectivity, and stability for formate production to BiNSs. The results showed that the selectivity of formate was 95.25% at −0.95 V (vs. RHE). Meanwhile, the N-BiNSs catalyst also showed excellent stability, and no apparent catalyst deactivation occurred after 20 h of electrolysis. Importantly, we also optimize the CO_2_RR activity by flow-cell. The N-BiNSs catalyst possessed high durability at a current density of 300 mA cm^−2^ at a potential of -1.2 V (vs. RHE), and the average FE of formate was about 90.30%. The DFT simulation suggested that compared with other catalysts, N-BiNSs had the active site to adsorb CO_2_ more easily. The free energy barrier for forming the critical intermediate *OCHO was reduced by nitrogen doping. This electrocatalyst with high catalytic activity, durability, and selectivity has great potential to improve the technical and economic feasibility of CO_2_ to formate conversion. In addition, this study provides a reasonable catalyst design for CO_2_RR studies by proposing a simple method for synthesizing nanostructured catalysts.

## Figures and Tables

**Figure 1 ijms-23-14485-f001:**
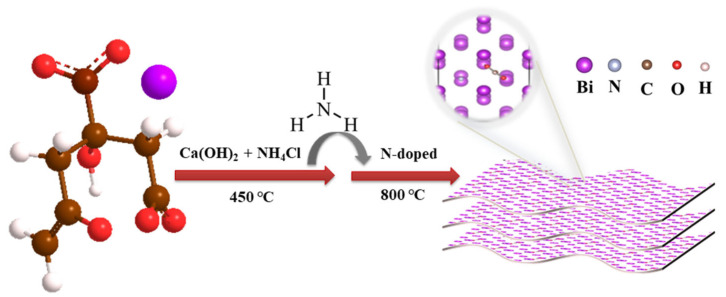
The illustration scheme of N-BiNSs preparation.

**Figure 2 ijms-23-14485-f002:**
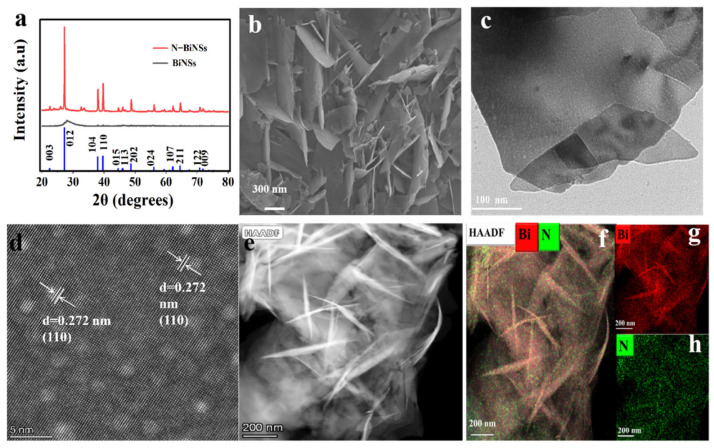
(**a**) XRD pattern of BiNSs and N-BiNSs. Characterization of N-BiNSs: (**b**) SEM image, (**c**) TEM image, (**d**) HR-TEM image, (**e**) HAADF-STEM and (**f**–**h**) EDS mapping (green and red represent N, Bi element, respectively).

**Figure 3 ijms-23-14485-f003:**
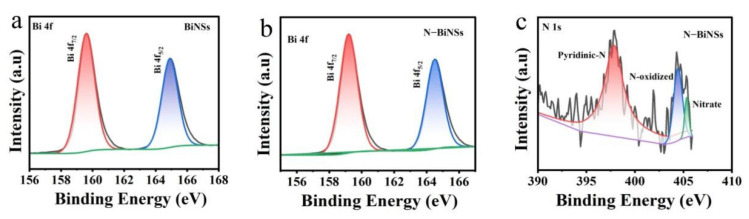
(**a**) XPS Bi 4f spectrum of BiNSs and (**b**) N-BiNSs; (**c**) XPS N 1s spectrum of N-BiNSs.

**Figure 4 ijms-23-14485-f004:**
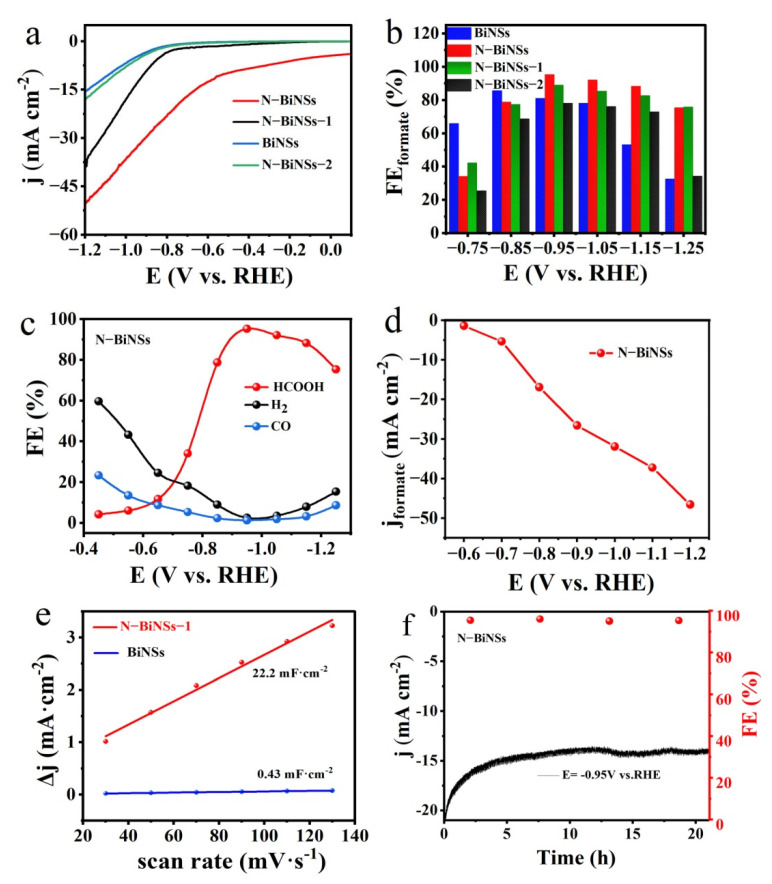
CO_2_RR performances of BiNSs, N-BiNSs, N-BiNSs-1, and N-BiNSs-2: (**a**) Linear sweep voltammetry curves in 0.5 M KHCO_3_ aqueous solutions with saturated gases CO_2_, sweeping speed of 5 mV s^−1^; (**b**) the product FE_formate_ at different applied potentials; (**c**) corresponding FE of N-BiNSs at various potentials; (**d**) the N-BiNSs of j_formate_ recorded at different potentials in 0.5 M KHCO_3_; (**e**) the BiNSs and N-BiNSs the relationship between charge current density difference (ΔJ) and scanning rate; (**f**) stability test of N-BiNSs in the H-type electrolytic cell and Faraday efficiency test of formate production.

**Figure 5 ijms-23-14485-f005:**
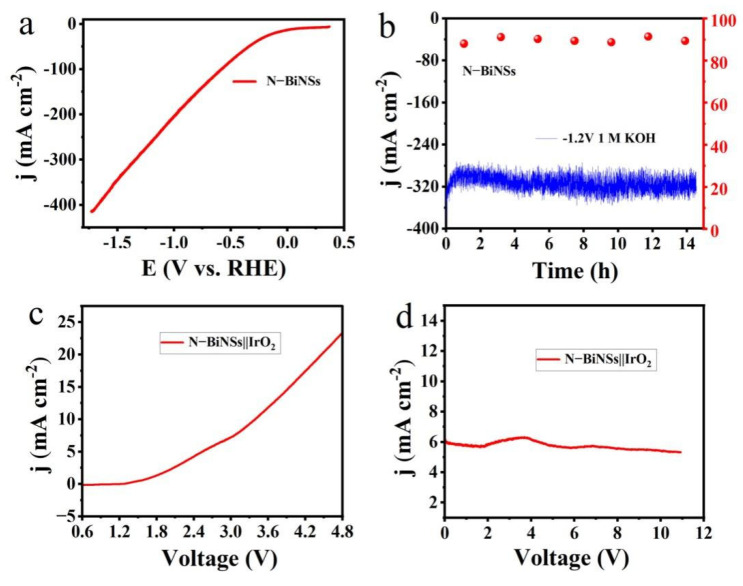
(**a**) The LSV curve in a flow cell under CO_2_ atmosphere; (**b**) Stability test of the material at −1.2 V vs. RHE and corresponding Faraday efficiency test of formate production of N-BiNSs; (**c**) Polarization curve of N-BiNSs||IrO_2_ couple in the two-electrode system; (**d**) Chronoamperometry measurement of N-BiNSs||IrO_2_ couple at 3 V.

**Figure 6 ijms-23-14485-f006:**
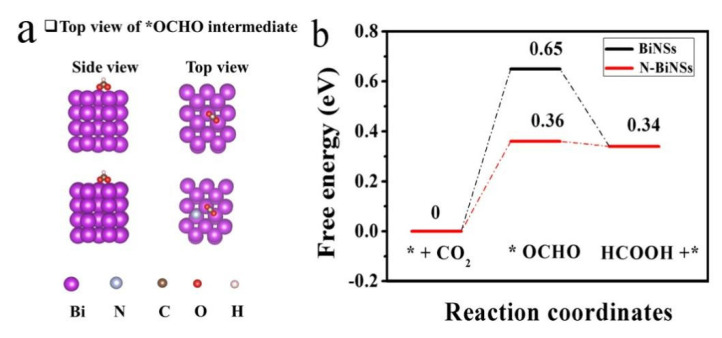
(**a**) Side and Top views of BiNSs (012) and N-BiNSs (012) configurations (the red, white, blue, purple and brown spheres represent: O, H, N, Bi, and C atoms, respectively); (**b**) Calculated free-energy diagram for *OCHO generation of N-BiNSs and BiNSs for the electrochemical reduction to formate process (where * represents the active site of the catalyst).

## Data Availability

All data in this study can be found in public data bases and Appendix A, as described in the Material and Methods section (Section 3).

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
