# Peer review of "Nitrogen-Doped Bismuth Nanosheet as an Efficient Electrocatalyst to CO2 Reduction for Production of Formate"

_ijms, 2022, doi:10.3390/ijms232214485_

Round 1

Reviewer 1 Report

The paper entitled "Nitrogen-doped Bismuth Nanosheets as an Efficient Electrocatalyst to CO2 Reduction for Production of Formate" (ijms-2042392) presents the synthesis of a bismuth-based electrocatalyst for the reduction of CO2 to formate. In particular, the authors studied the morphological characteristics of the catalyst, the electrochemical performance and present a study on the mechanism of the reaction through DFT calculations

The work would seem interesting and suitable for publication with some small revisions, listed below:

1) there are two figure 4

2) in the figure 3 reference is made to the signals of 4f (7/2) and 4f (5/2), but in the text there is only 4f (5/2)

3) the meaning of several sentences is not clear. They must be rewritten to allow a better understanding of the text (for example, line 54, line 66, line 149, lines 157-158, lines 167-168)

4) lines 178: the text refers to the inset of the figure. This is really small. I suggest that the authors separate it from the others or bring it to the supplementary file

5) figure S5b: it would be better for the impedances to have the same dimension on the two axes

6) line 232: the statement " The OER electrocatalyst IrO2 may also be replaced by non-noble metal-based materials" should be accompanied by some reference

Author Response

  1. There are two figure 4.

Response: Thank you very much for your thoughtful questions. We carefully checked the figure notes and renumbered the figures in the revised version.(Revised manuscript)

  1. In the figure 3 reference is made to the signals of 4f (7/2) and 4f (5/2), but in the text there is only 4f (5/2).

Response: Thank you very much for your valuable questions. In the revised version, we have carefully modified the expression in the article, corresponding to  the information in the figure. “For the BiNSs catalyst, binding energies at 165 eV and 159.6 eV were belong to the Bi3+ 4f5/2 and 4f7/2 [23], respectively. However, binding energies for Bi3+ 4f5/2 and 4f7/2 in N-BiNSs shifted to 164.5 and 159.11 eV, respectively. ”(Page 3, Revised manuscript)

  1. The meaning of several sentences is not clear.They must be rewritten to allow a better understanding of the text (for example, line 54, line 66, line 149, lines 157-158, lines 167-168).

Response: Thank you for the valuable comment. According to your suggestion, we have revised the sentences in the revised version. (Revised manuscript)

  1. Lines 178: the text refers to the inset of the figure. This is really small.I suggest that the authors separate it from the others or bring it to the supplementary file.

Response: According to your suggestion, we had modified Figure 4e and transferred the inset figure to the supplementary information (Figure S5c Supplementary information).

  1. Figure S5b: it would be better for the impedances to have the same dimension on the two axes.

Response: According to your suggestion, in the revised version, we modified the dimensions on both axes in Figure S5b. (Supplementary information)

  1. 6.Line 232: the statement " The OER electrocatalyst IrO2may also be replaced by non-noble metal-based materials" should be accompanied by some reference.

Response: Thank you very much for your valuable comments. We have cited the reference (ref. 37) in the revised version. “The OER electrocatalyst IrO2 may also be replaced by non-noble metal-based materials [37]” (page 6).

Reviewer 2 Report

Recommendation: Minor revision

General comments:

The present manuscript reported a nitrogen-doped bismuth nanosheets (N-BiNSs)  catalyst with efficient electrocatalytic performance for CO2RR with selective formate production. In H-type electrolytic cell, the maximum FEformate at −0.95V (vs. RHE) reaches 95.25.0% in 0.5 M KHCO3 electrolyte. In addition, the N-BiNSs for CO2RR yielded large current density (300 mA cm-2) for fomate production in a flow-cell system. The manuscript is well organized and the experimental results can support their conclusions. However, there are also some problems with this article. Therefore, I recommend its publication in this journal after minor revision.

1. Page 8, in experimental section, in describing the process of catalyst synthesis, it is suggested that the athuors should use the reagents should be described into the corresponding chemical formula for all reagents, such as “ammonium chloride” can be written as NH4Cl.

2.  In the article, there are a few spelling mistakes and incorrect use of word parts of speech, such as, assebbleelectrocatalytic”. Meanwhile, there are also some grammatical mistakes in the article. Please check and correct them carefully.

3. Are there some changes for the N-doped Bi in the electrolysis process?

Author Response

  1. 1.Page 8, in experimental section, in describing the process of catalyst synthesis, it is suggested that the athuors should use the reagents should be described into the corresponding chemical formula for all reagents, such as “ammonium chloride” can be written as NH4Cl.

Response: Thank you very much for your valuable comments. According to your comments, we used the corresponding chemical formula in experimental section for the reagents in the revised version. (Revised manuscript)

  1. 2.In the article, there are a few spelling mistakes and incorrect use of word parts of speech, such as, “assebble”、“electrocatalytic”.  Meanwhile, there are also some grammatical mistakes in the article.  Please check and correct them carefully.

Response: Thank you very much for your valuable comments. We carefully correct a  spelling mistakes in the revised version. Besides, we have corrected the syntax errors in the revised version. (Revised manuscript)

  1. 3.Are there some changes for the N-doped Bi in the electrolysis process?.

Response: The N-BiNSs catalyst showed that the morphology of the nanosheets was maintained and became thinner after a period of electrolysis process (Figure S7a). The crystal shape of the nanosheets remained after prolonged electrolysis (Figure S7b). It is proved that the catalyst had excellent morphology stability.
